USTC-ICTS/PCFT-24-02

# Partial entanglement network and bulk geometry reconstruction in AdS/CFT

Jiong Lin[1,2,3] *, Yizhou Lu[4] ⋆ and Qiang Wen[5] †

**1** School of Physics and Information Engineering, Guangdong University of Education, Guangzhou 510303, China
**2** Interdisciplinary Center for Theoretical Study, University of Science and Technology of China, Hefei, Anhui 230026, China
**3** Peng Huanwu Center for Fundamental Theory, Hefei, Anhui 230026, China
**4** Department of Physics, Southern University of Science and Technology, Shenzhen 518055, China
**5** Shing-Tung Yau Center and School of Physics, Southeast University, Nanjing 210096, China
E-mail: * jionglin@ustc.edu.cn, ⋆ luyz@sustech.edu.cn, † wenqiang@seu.edu.cn

## Abstract

In the context of Anti-de Sitter / Conformal Field Theory (AdS/CFT) correspondence, we present a general scheme to reconstruct bulk geometric quantities in static AdS background with the partial entanglement entropy (PEE), which is a measure of the entanglement structure on the boundary CFT. The PEE between any two points $\mathcal{I}(\mathbf{x}, \mathbf{y})$ is the fundamental building block of the PEE structure. Following [1], we geometrize any two-point PEE $\mathcal{I}(\mathbf{x}, \mathbf{y})$ into the bulk geodesic connecting the two boundary points $\mathbf{x}$ and $\mathbf{y}$, which we refer to as the PEE thread. Thus, in the AdS bulk we get a continues "network" of the PEE threads, with the density of the threads determined by the boundary PEE structure. In this paper, we demonstrate that the strength of the PEE flux at any bulk point along any direction is $1/4G$. This observation give us a reformulation for the RT formula. More explicitly, for any static boundary region $A$ the homologous surface $\Sigma_A$ that has the minimal flux of the PEE threads passing through it is exactly the Ryu-Takayanagi (RT) surface of $A$, and the minimal flux coincides with the holographic entanglement entropy of $A$. Furthermore, we demonstrate that any geometric quantities can be reconstructed by the PEE threads passing through it, which can further be interpreted as an integration of the boundary two-point PEEs

# 1   Introduction

The AdS/CFT correspondence [2–4] states that the quantum theory of gravity in asymptotic AdS$_{d+1}$ spacetime is equivalent to a certain CFT$_d$ on the asymptotic boundary. This provides a window to understand both of the classical and quantum aspects of gravitational theories based on the information in the boundary CFT, using the dictionary of the correspondence. Several important developments have been made along this line [5–12], where the insights from a holographic perspective of boundary quantum entanglement structure have played a central role. These achievements began with the Ryu-Takayanagi (RT) formula [5] which relates the entanglement entropy of any boundary region to the area of the bulk minimal surface homologous to that boundary region. This proposal was later refined to the covariant version [6,12] and the version including quantum corrections [7,8,10,11]. For more recent developments on the bulk reconstruction inspired by holographic study of quantum entanglement, one should consult the following review papers [13, 14].

The possibility to reconstruct the bulk geometry from the entanglement structure of the boundary field theory was soon realized after the RT formula was proposed, see [15,16] for the earliest discussion. In this paper, we focus on the reconstruction of bulk geometric quantities in terms of boundary entanglement structure measures. So far, several approaches have been explored for this goal. For example, the reconstruction of certain bulk curves via the *differential entropy* [17–22] by studying the geodesics tangent to the curve, the reformulation of the RT formula as the maximal flux of the *bit threads* in AdS space out of the region [23–25], and the simulation of the AdS space based on the tensor networks [26–34] where the RT surface is interpreted as the homologous path in the network with the minimal number of cut. See [35] for a detailed review on the above approaches.

Nevertheless, the geometry reconstruction program is far from complete. The *differential entropy* scheme works clearly in reconstructing certain type of curves in AdS$_3$ and it turns out to be messy in higher dimensions [36]. The bit threads scheme works fine for static configurations in general dimensions, but it is not clear how to reconstruct geometric quantities beyond the RT surfaces via the bit threads. Also, the bit thread configuration depends on the choice of boundary regions we study, and is highly degenerate even with a given region. Hence it is not clear what we can learn from an explicit configuration of bit threads beyond the holographic entanglement entropy. The tensor networks are a toy models that can reproduce some of the important features of AdS/CFT, including AdS background geometry (see [37] for an exploration), the RT formula for holographic entanglement entropy [26, 32], the quantum error correction of holography [31] and so on. Nevertheless, it is also subtle or hard to extend the simulation of AdS/CFT via tensor networks to higher dimensions and time dependent configurations. Also, the interpretation for geometric quantities beyond the RT surface in terms of the tensor network structure is rarely explored.

Inspired by these approaches, we propose a framework to reconstruct all the bulk geometric quantities based on a new measure of entanglement, the *partial entanglement entropy* (PEE) [38–42]. See [43–52] for discussions on the relation between PEE and the above three approaches, and see also [53–58] for the reconstruction of the entanglement wedge cross-section in various scenarios based on PEE. Our scheme works quite clearly in reconstructing

a generic geometric quantity (or any co-dimension two surfaces) in static AdS bulk in general dimensions.

In section 2, we will first introduce PEE in the vacuum state of a holographic CFT, its geometrization as the PEE threads in the bulk, and the weight of the PEE threads. In section 3, we use the PEE threads to give a reformulation of the RT formula, in a way quite similar to the calculation of the entanglement entropy using tensor networks. In section 4, we use the PEE threads to reconstruct a generic bulk co-dimension two surface. In the last section, we give a conclusion and discussion for the results.

## 2 The partial entanglement network

### 2.1 Partial entanglement entropy

The PEE $\mathcal{I}(A, B)$ is a special measure of two-body correlation between two non-overlapping regions $A$ and $B$ [38,39,41,42]. Besides all the physical properties that are satisfied by mutual information $I(A, B)$, PEE possesses an exclusive property of additivity [39,59]. More explicitly, assuming that $A$, $B$ and $C$ are three non-overlapping regions, the physical requirements for the PEE are classified in the following

1. **Additivity:** $\mathcal{I}(A, B \cup C) = \mathcal{I}(A, B) + \mathcal{I}(A, C)$;

2. **Permutation symmetry:** $\mathcal{I}(A, B) = \mathcal{I}(B, A)$;

3. **Normalization:** $\mathcal{I}(A, B)|_{B \to \bar{A}} = S_A$;

4. **Positivity:** $\mathcal{I}(A, B) > 0$;

5. **Upper bounded:** $\mathcal{I}(A, B) \le \min\{S_A, S_B\}$;

6. $\mathcal{I}(A, B)$ should be **invariant under local unitary transformations** inside $A$ or $B$;

7. **Symmetry:** for any symmetry transformation $\mathcal{T}$ under which $\mathcal{T}A = A'$ and $\mathcal{T}B = B'$, we have $\mathcal{I}(A, B) = \mathcal{I}(A', B')$.

As was shown in [39,60], the above requirements has q unique solution for states with Poncaré symmetry. Furthermore, for the vacuum state of a CFT on a plane, the formula of the solution is totally determined by the above requirements.

According to the properties of additivity and permutation symmetry, the PEE structure are fully described by the two-point PEEs $I(\mathbf{x}, \mathbf{y})$, and $\mathcal{I}(A, B)$ can be written as a double integral over $A$ and $B$

$$\mathcal{I}(A, B) = \int_A d\sigma_{\mathbf{x}} \int_B d\sigma_{\mathbf{y}} \, \mathcal{I}(\mathbf{x}, \mathbf{y}). \tag{1}$$

where $\sigma_{\mathbf{x}, \mathbf{y}}$ are the infinitesimal area element at $\mathbf{x}$ and $\mathbf{y}$, and the two-point PEE $\mathcal{I}(\mathbf{x}, \mathbf{y})$ in vacuum $\text{CFT}_d$ is given by

$$\mathcal{I}(\mathbf{x}, \mathbf{y}) = \frac{c}{6} \frac{2^{d-1}(d-1)}{\Omega_{d-2}|\mathbf{x} - \mathbf{y}|^{2(d-1)}}, \tag{2}$$

where $\Omega_{d-2} = 2\pi^{\frac{d-1}{2}}/\Gamma\left(\frac{d-1}{2}\right)$ is the area of $(d-2)$-dimensional unit sphere. One can either derive the above formula for two-point PEE via the solution to all the physical requirements [39, 60], or using the so-called additive-linear-combination (ALC) proposal in quasi-one-dimensional system to construct PEE [41]. See [1] for the details about the derivation of (2).

The normalization property of the PEE $\mathcal{I}(A,B)|_{B\to\bar{A}} = S_A$ tells us how to approach the entanglement entropy from PEE,

$$S_A = \int_A \mathrm{d}\sigma_{\mathbf{x}} \int_{\bar{A}-\epsilon} \mathrm{d}\sigma_{\mathbf{y}}\, \mathcal{I}(\mathbf{x},\mathbf{y}), \tag{3}$$

where $A \cup \bar{A}$ makes up a pure state and $\epsilon$ represents a regularization cutoff. Note that, the requirement of normalization is quite subtle as it is an equality between two divergent quantities which are normalized in different schemes, see [41] for more discussions on this requirement. We should keep in mind that, we only impose the normalization requirement to spherical regions, where the relation between the geometrical cutoff and the UV cutoff can be explored, to get the solution (2). The solution may not exist if we impose the normalization requirement to generic regions (especially disconnected regions)[1]. As was implied in [1], the modification of this requirement for generic regions is the key for our scheme to reconstruct the geometric quantities in AdS. Although the PEE structure (2) may not capture all the information of the entanglement structure in a CFT, we will see that it is enough to reconstruct the geometric quantities at order $\mathcal{O}(c)$ in the gravity side of AdS/CFT.

## 2.2 PEE threads and the partial entanglement network

In AdS/CFT, we introduced a scheme to geometrize the PEEs in [1], where the boundary two-point PEEs $\mathcal{I}(\mathbf{x},\mathbf{y})$ are represented by the bulk geodesics connecting the two boundary points $\{\mathbf{x},\mathbf{y}\}$, which we call the PEE *threads* [1]. This geometrization looks quite like the bit thread configurations. Nevertheless they are quite different objects, for example the PEE thread configuration is totally determined by the boundary state and they intersect with each other, while the bit thread configuration are highly degenerate and bit threads do not intersect with each other. We only consider the vacuum state of the $\text{CFT}_d$ and a static time slice in $\text{AdS}_{d+1}$. The PEE *threads* emanating from any point $\mathbf{x}$ can be represented by a divergenceless vector field $V_{\mathbf{x}}^{\mu} = |V_{\mathbf{x}}|\tau^{\mu}$, where $\tau^{\mu}$ is the unite vector tangent to the geodesics emanating from $\mathbf{x}$. The norm $|V_{\mathbf{x}}|$ characterizes the density of the threads, which is determined by the requirement that,

- *the flux of the PEE* threads *from $\mathbf{x}$ to any boundary region* $\mathrm{d}\sigma$ *should match the PEE* $\mathcal{I}(\mathbf{x},\mathrm{d}\sigma)$.

In summary, given the PEE structure of the boundary CFT and the metric of the dual spacetime, we get a network of the PEE *threads* in the AdS bulk consisting of all the bulk geodesics on a time slice anchored on the boundary (see Fig. 1 for examples). We call it the *partial entanglement network*, or *PEE network* for short.

We briefly review the derivation of the PEE vector flow $V_{\mathbf{x}}^{\mu}$ and its connection with the configuration of bit threads [1]. Due to the translation symmetry, it is sufficient to derive the PEE threads emanating from the origin, $V_O^{\mu} \equiv V_{\mathbf{x}=0}^{\mu}$. We work in poincaré $\text{AdS}_{d+1}$ with unit AdS radius,

$$\mathrm{d}s^2 = \frac{-\mathrm{d}t^2 + \mathrm{d}z^2 + \mathrm{d}r^2 + r^2\mathrm{d}\Omega_{d-2}^2}{z^2}, \tag{4}$$

---

[1]Nevertheless, in $\text{CFT}_d$ entanglement entropies for various shapes of connected regions have been carried out based on (2) and (3) [61–64], which are in good (but not exact) agreement with the results derived from other methods. In these papers, the authors studied the mutual information that satisfies additivity (EMI), which coincides with the PEE [39] in these scenarios. See [39, 41] for discussion on the relationship between the PEE and the EMI.

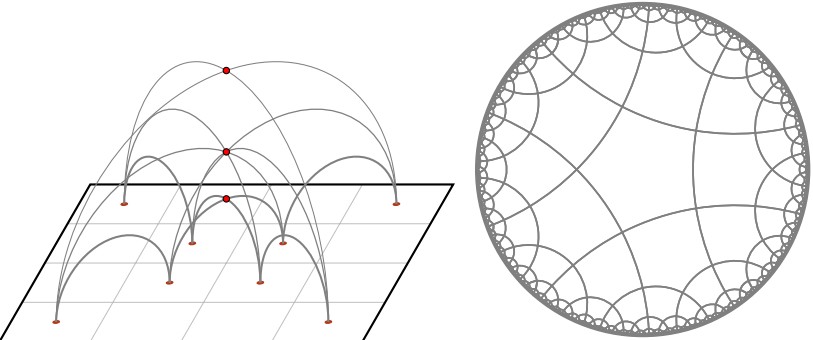

Figure 1: Visualizations of PEE threads (gray) on a time slice of Poincaré AdS$_4$ and global AdS$_3$, where the boundary sites (on the plane) are discretized. Left: the PEE threads with larger radii are denoted with thinner curves, whose thickness reflects the density of the PEE threads. Right: with appropriate discretizations, they form a hyperbolic tiling.

where

$$d\Omega_{d-2}^2 = d\phi_1^2 + \sin^2\phi_1 d\phi_2^2 + \cdots + \sin^2\phi_1 \cdots \sin^2\phi_{d-3} d\phi_{d-2}^2. \tag{5}$$

In higher dimensions, due to the rotational symmetry of $V_O^\mu$, we can restrict to a 2-dimensional slice with $\phi_i = 0$. Since the PEE threads are just the bulk geodesics emanating from $O$, the vector field $V_O^\mu(Q)$ is tangent to these geodesics, such that

$$V_O^\mu(Q) = |V_O(Q)|\tau_O^\mu(Q), \tag{6}$$

where

$$\tau_O^\mu(Q) = \frac{2zr}{r^2 + z^2}\left(z, \frac{z^2 - r^2}{2r}\right), \tag{7}$$

is the unit vector tangent to the geodesics emanating from $O$, and $z$, $r$ are the coordinates of the bulk point $Q$ on the $\phi_i = 0$ slice. The norm $|V_O(Q)|$ is then settled down by the requirement that, the flux of the PEE threads from the origin to any boundary region $dy$ should match the PEE $\mathcal{I}(0, dy)$.

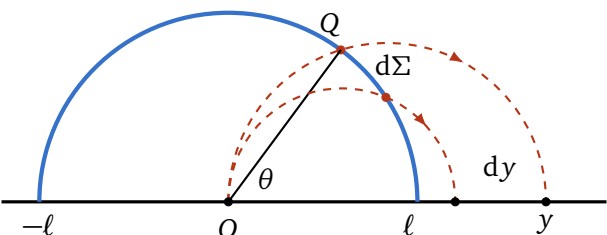

Figure 2: Here $d\Sigma$ is an infinitesimal area element at $Q$, and the blue circle is the reference RT surface $r^2 + z^2 = \ell^2$. The PEE threads emanating from $O$ and passing through $d\Sigma$ will anchor on a boundary region $dy$. There exists an one-to-one mapping between any point $(l\cos\theta, l\cos\theta)$ on $d\Sigma$ and the point $y = \ell/\cos\theta$ in the $dy$ region. This pair of points are connected by a PEE thread emanated from $O$.

More explicitly, let us consider the coordinate $Q = (\bar{r}, \bar{z}) = (\ell\cos\theta, \ell\sin\theta)$ and a reference RT surface $\Sigma$ passing through $Q$ with the radius $\ell = \sqrt{r^2 + z^2}$ (see Fig. 2). The flux of the PEE

threads $V_O^\mu$ through an area element $d\Sigma$ at $Q$ on $\Sigma$ is given by

$$\begin{aligned}
\text{Flux}(V_O^\mu, d\Sigma) &= d\theta \, d\Omega_{d-2} \sqrt{h} V_O^\mu(\theta) n_{\Sigma,\mu}(\theta) \\
&= d\theta \, d\Omega_{d-2} \frac{\bar{r}^{d-2}}{\bar{z}^{d-2}} \sqrt{h_{\theta\theta}} |V_O(\theta)| \sin\theta \\
&= d\theta \, d\Omega_{d-2} \sqrt{h_{\theta\theta}} |V_O(\theta)| \cot^{d-2}\theta \sin\theta,
\end{aligned}$$ (8)

where $\theta = \arctan\frac{z}{r}$, $h_{\theta\theta} = 1/\sin^2\theta$ is the $\theta\theta$-component of the induced metric on $\Sigma$ and $n_\Sigma^\mu(Q) = \ell \sin\theta(\cos\theta, \sin\theta)$ is the unit normal vector on $\Sigma$. We have used $\sqrt{h} = \frac{\bar{r}^{d-2}}{\bar{z}^{d-2}} \sqrt{h_{\theta\theta}}$. On the other hand the PEE $\mathcal{I}(O, dy)$ is given by $y^{d-2} \mathcal{I}(0, y) \Omega_{d-2} dy$. Then we solve the requirement

$$\text{Flux}(V_O^\mu, d\Sigma) = dy \, y^{d-2} \mathcal{I}(0, y) \Omega_{d-2}, \quad y = \ell/\cos\theta$$ (9)

to get the norm of $V_O(Q)$,

$$|V_O(Q)| = \frac{1}{4G} \frac{2^{d-1}(d-1)}{\Omega_{d-2}} \frac{z^{d-1}}{(r^2 + z^2)^{d-1}}.$$ (10)

In the above derivation we used the mapping $y = \ell/\cos\theta$ since $d\Sigma$ determines the $dy$ region on the boundary following the PEE threads. Finally, we obtain the PEE threads flow on the $\phi_i = 0$ slice

$$V_O(Q) = \frac{2^d z^d}{4G} \frac{(d-1)}{\Omega_{d-2}} \frac{r}{(r^2 + z^2)^d} \left(z, \frac{z^2 - r^2}{2r}, 0, \cdots\right).$$ (11)

Due to the translation symmetry, the vector flow from any boundary point $\mathbf{x}$ is identical to (11) up to a translation. For example, in AdS$_3$, the PEE thread flow emanating from $r = r_0$ can be obtained by replacing $r$ with $r - r_0$ in (11) and

$$V_{r_0}^\mu = \frac{1}{4G} \frac{2z^2(r - r_0)}{((r - r_0)^2 + z^2)^2} \left(z, \frac{z^2 - (r - r_0)^2}{2(r - r_0)}\right).$$ (12)

Having determined the PEE flow from any boundary point, now we sum up the PEE threads emanating from the points inside a sperical region $A = \{\mathbf{x} | |\mathbf{x}| < R\}$, to get a new vector flow

$$V_A^\mu = \int_A d^{d-1}\mathbf{x} V_{\mathbf{x}}^\mu = \frac{1}{4G} \left(\frac{2Rz}{\sqrt{(R^2 + r^2 + z^2)^2 - 4R^2 r^2}}\right)^d \left(\frac{rz}{R}, \frac{R^2 - r^2 + z^2}{2R}\right),$$ (13)

which is exactly the bit thread [23] configuration for $A$ constructed in [65]. In this configuration, all the bit threads flow along geodesics normal to the RT surface. Hence this bit thread flow is just a superposition of the PEE flow emanating from all the points in the region under consideration. Which is consistent with the fact that the bit thread flow depend on the choice of the region. One can check that the PEE (thread) flux at any point $Q$ on the RT surface $\mathcal{E}_A$ of $A$ is just

$$\int_A V_{\mathbf{x}}^\mu(Q) d^{d-1}\mathbf{x} = \frac{1}{4G} n^\mu(Q),$$ (14)

whose strength is a constant $1/(4G)$, which is independent from the radius and position of $A$. Here $n^\mu(Q)$ is the outward normal unit vector on the RT surface of $A$.

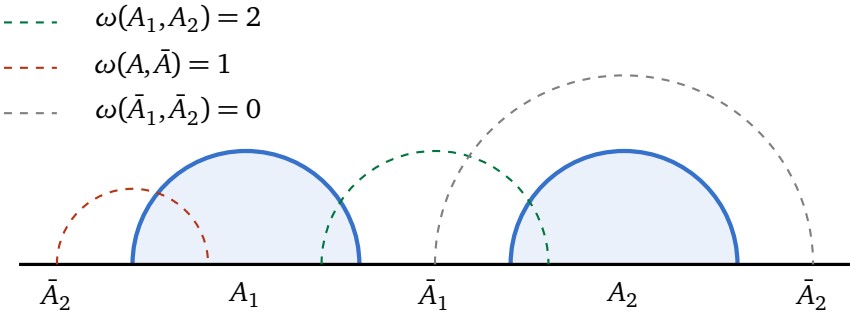

Figure 3: This figure is extracted from [1]. Here $A = A_1 \cup A_2$, $\bar{A} = \bar{A}_1 \cup \bar{A}_2$ and $A\bar{A}$ is in the vacuum state of the holographic $CFT_2$. Representative PEE threads (dashed curves) in Poincaré $AdS_3$. Each PEE thread crosses the disconnected RT surface for different number of times, which represents different weight.

## 2.3 The weight of the PEE threads

Now we try to calculate the entanglement entropy using the PEE threads. If we naively apply the normalization property to any region $A$, then the entanglement entropy is calculated by (3), which has a corresponding picture in the partial entanglement network. This equates to calculating the flux of the PEE thread flow from $A$ to $\bar{A}$

$$S_A = \int_{\Sigma_A} \mathrm{d}\Sigma_A \sqrt{h} V_A^\mu n_\mu, \tag{15}$$

where $\Sigma_A$ is any co-dimension two surface homologous to $A$, $n^\mu$ is the outward normal unit vector on $\Sigma_A$ and $h$ is the induced metric. In this paper, we denote $\Sigma_A$ as any homologous surface to $A$, and $\mathcal{E}_A$ as the RT surface of $A$, that is $\mathrm{Area}[\mathcal{E}_A] = \min \mathrm{Area}[\Sigma_A]$. Naively, we expect that (15) should reproduce the same entanglement entropy as the RT formula for an arbitrary region $A$. From (14), we could deduce that this holds for any spherical region. Nevertheless, this coincidence does not happen for other regions, see [1] for case of a strip region.

To fix this problem, we should modify (3) in some way. The discussion for the two-interval case in [1] in $CFT_2$ provides a clear clue. Let us consider a two interval region $A = A_1 \cup A_2$ whose RT surface is given by the blue curves, see Fig. 3. If we naively apply (3), we should only count the threads connecting $A_i$ and $\bar{A}_j$, which gives $S_A = \mathcal{I}(A_1, \bar{A}) + \mathcal{I}(A_2, \bar{A})$. However, the RT formula implies

$$\begin{aligned} S_A &= S_{A_1} + S_{A_2} = \mathcal{I}(A_1, A_2\bar{A}) + \mathcal{I}(A_2, A_1\bar{A}) \\ &= \mathcal{I}(A_1, \bar{A}) + \mathcal{I}(A_2, \bar{A}) + 2\mathcal{I}(A_1, A_2), \end{aligned} \tag{16}$$

where we used (3) for $A_1$ and $A_2$ as it applies to single intervals. So, (16) indicates that the RT formula not only counts the threads connecting $A$ and $\bar{A}$, but also doubly counts the threads connecting $A_1$ and $A_2$. This absolutely goes beyond (3), but looks reasonable as the threads connecting $A_1$ and $A_2$ intersect with the RT surface twice.

Then one may weight the threads with the number of times it intersects with the RT surface. For any disjoint multi-interval $A = \cup A_i$ and its complement $\bar{A} = \cup \bar{A}_j$, given the RT surface we can read out the weight for any PEE threads. Let us denote the weight for the threads connecting any two sub-intervals $\alpha$ and $\beta$ as $\omega(\alpha, \beta)$. It has further been checked in [1] that the holographic entanglement entropy is given by

$$S_A = \sum_{i,j} \left( \omega(A_i, A_j)\mathcal{I}(A_i, A_j) + \omega(\bar{A}_i, \bar{A}_j)\mathcal{I}(\bar{A}_i, \bar{A}_j) + \omega(A_i, \bar{A}_j)\mathcal{I}(A_i, \bar{A}_j) \right). \tag{17}$$

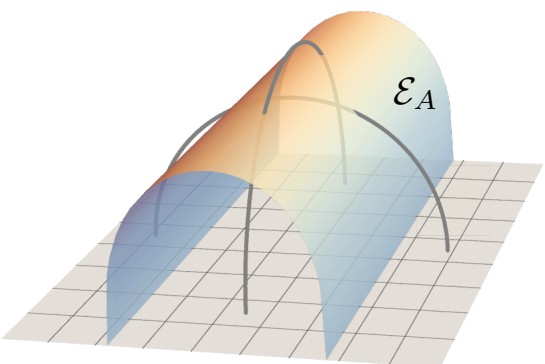

Figure 4: The RT surface $\mathcal{E}_A$ for a strip region $A$ in Poincaré AdS$_4$. The PEE *threads* (the gray curves) with both endpoints inside or outside $A$ can possibly pass through $\mathcal{E}_A$ twice.

The above observation is consistent with (3) for single interval cases where only the threads connecting the interval $A$ and its complement $\bar{A}$ have non-zero weight $\omega(A, \bar{A}) = 1$. From bulk PEE flux perspective, it is important to note that, (17) indeed computes the PEE flux that flows from the entanglement wedge $\mathcal{W}_A$ to $\mathcal{W}_{\bar{A}}$, instead of the flux from $A$ to $\bar{A}$.

## 3   Reformulation of the RT formula

Now we extend to general dimensional Poincaré AdS$_{d+1}$ and get rid of the precondition that we know the RT surface. For any disconnected boundary region $A = \cup A_i$ and its compliment $\bar{A} = \cup \bar{A}_j$, we consider an arbitrary co-dimension two surface $\Sigma_A$ homologous to $A$. If we define the weight of a PEE thread as the number it intersects with the surface $\Sigma_A$, then the configuration of the weights depends on $\Sigma_A$. In higher dimensions, PEE threads that anchored on the same connected subregion $A_i$ can pass through the RT surface hence has non-zero weight (see Fig. 4 for the example of a strip region). So, when we talk about the weight of a PEE thread, we should also specify the two boundary points it connects. For these reasons, we denote the weight of the thread connecting the pair of the boundary points $\{\mathbf{x}, \mathbf{y}\}$ as $\omega_{\Sigma_A}(\mathbf{x}, \mathbf{y})$.

Inspired by the above observation (17), here we give a proposal as a complete reformulation of the RT formula. Given a homologous surface $\Sigma_A$, it always divides the bulk space $\mathcal{M}$ into two parts $\mathcal{M}_A$ and $\mathcal{M}_{\bar{A}}$ whose boundaries satisfy

$$\partial \mathcal{M}_A = A \cup \Sigma_A, \qquad \partial \mathcal{M}_{\bar{A}} = \bar{A} \cup \Sigma_A. \tag{18}$$

We propose that, the $\Sigma_A$ that minimizes the PEE flux from $\mathcal{M}_A$ to $\mathcal{M}_{\bar{A}}$ is exactly the RT surface, and the corresponding minimal flux coincides with the holographic entanglement entropy, i.e.

$$S_A = \min_{\Sigma_A} \frac{1}{2} \int_{\partial \mathcal{M}} \mathrm{d}^{d-1}\mathbf{x} \int_{\partial \mathcal{M}} \mathrm{d}^{d-1}\mathbf{y} \, \omega_{\Sigma_A}(\mathbf{x}, \mathbf{y}) \mathcal{I}(\mathbf{x}, \mathbf{y}), \tag{19}$$

where the integration domain of $\mathbf{x}$ and $\mathbf{y}$ is the whole AdS boundary $\partial \mathcal{M} = A\bar{A}$. Here we have considered all the PEE threads and their weights instead of only those connecting $A$ and $\bar{A}$. The factor $1/2$ appears as we count both $\mathcal{I}(\mathbf{x}, \mathbf{y})$ and $\mathcal{I}(\mathbf{y}, \mathbf{x})$ in the integration. From the bulk perspective, the above equation can also be written in terms of the PEE vector flows

$$S_A = \min_{\Sigma_A} \frac{1}{2} \int_{\Sigma_A} \mathrm{d}\Sigma_A \sqrt{h} \int_{\partial \mathcal{M}} \mathrm{d}^{d-1}\mathbf{x} |V_{\mathbf{x}}^{\mu}(Q) n_{\mu}(Q)|, \tag{20}$$

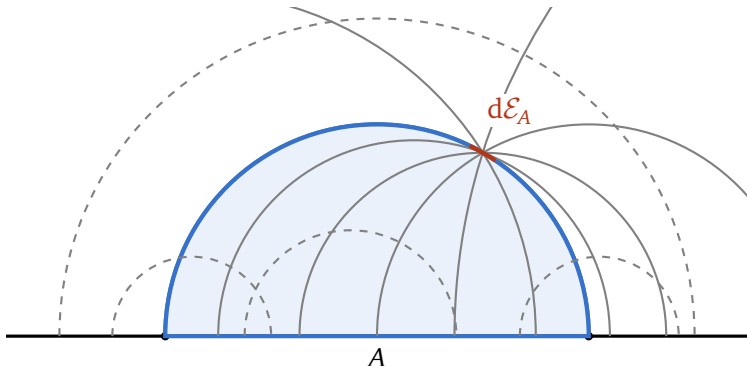

Figure 5: Representative PEE threads in Poincaré AdS$_3$. The blue shaded region is the entanglement wedge $\mathcal{W}_A$ of $A$. PEE threads that pass through a surface segment $\mathrm{d}\mathcal{E}_A$ of the RT surface are denoted by gray solid curves. PEE threads that do not intersect $\mathrm{d}\mathcal{E}_A$ are denoted by gray dashed curves.

where $Q$ is any point on $\Sigma_A$, $n^\mu$ is the unit normal vector at $Q$ pointing from $\mathcal{M}_A$ to $\mathcal{M}_{\bar{A}}$ and $\mathrm{d}\Sigma_A\sqrt{h}$ is the area of an infinitesimal area element on $\Sigma_A$ located at $Q$. Similarly, the coefficient $1/2$ appears because we integrate $\mathbf{x}$ over the whole boundary such that the PEE thread connecting any two boundary point is doubly counted. Here we take the absolute value for $V_{\mathbf{x}}^\mu n_\mu$ since locally we are always calculating the flow from one side of $\Sigma_A$ to the other side, hence any PEE thread passing through $\Sigma_A$ gives positive contribution. The equivalence between (20) and (19) is guaranteed by the divergenceless property of the vector fields $V_{\mathbf{x}}^\mu$.

The version of the above proposal (19) (or (20)) in AdS$_3$/CFT$_2$ was proposed and tested in [1] without a proof. In the following we present simple and general proof for any static boundary region $A$ in AdS$_{d+1}$/CFT$_d$. This proposal gives a complete reformulation of the RT formula, as the minimization reproduces the RT surface, and the minimized flux reproduces the holographic entanglement entropy.

**Proof**

Although the analysis of the PEE flow for static spherical regions [1] looks quite special, it contains the key ingredient to prove our proposal for generic regions. Let us consider a spherical region $A$, whose RT surface $\mathcal{E}_A$ is just a hemisphere, and all the PEE threads can be classified into three classes,

1.  $\omega_{\mathcal{E}_A}(\mathbf{x}, \mathbf{y}) = 0$ for $\mathbf{x}, \mathbf{y} \in A$;

2.  $\omega_{\mathcal{E}_A}(\mathbf{x}, \mathbf{y}) = 0$ for $\mathbf{x}, \mathbf{y} \in \bar{A}$;

3.  $\omega_{\mathcal{E}_A}(\mathbf{x}, \mathbf{y}) = 1$ for $\mathbf{x} \in A, \mathbf{y} \in \bar{A}$ or $\mathbf{x} \in \bar{A}, \mathbf{y} \in A$.

Only the threads in the third class intersect with $\mathcal{E}_A$. More specifically, let us consider an infinitesimal area element $\sqrt{h}\mathrm{d}\mathcal{E}_A$ on the RT surface. From any boundary point, there are PEE threads passing through $\sqrt{h}\mathrm{d}\mathcal{E}_A$. In order to get the right PEE flux through $\sqrt{h}\mathrm{d}\mathcal{E}_A$, the key is to avoid double counting. It is obvious that we only need to sum over the threads emanating from $A$ (see the gray solid curves Fig. 5) to get all the threads passing through $\sqrt{h}\mathrm{d}\mathcal{E}_A$ without double counting. So we only need to integrate the PEE flow $V_{\mathbf{x}}^\mu$ for $\mathbf{x} \in A$ to get the flux of PEE threads passing through the area element, which is just (14)

$$\sqrt{h}\mathrm{d}\mathcal{E}_A \int_A \mathrm{d}^{d-1}\mathbf{x} \, V_{\mathbf{x}}^\mu n_\mu = \frac{\sqrt{h}\mathrm{d}\mathcal{E}_A}{4G}. \tag{21}$$

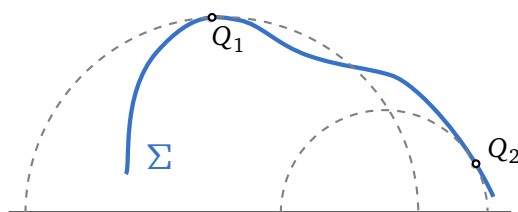

Figure 6: Here $\Sigma$ represents an arbitrary codimension-2 bulk curve in AdS$_3$. Any infinitesimal line segment at $Q_i$ is reconstructed by the set of PEE threads passing through it and emanating from the spherical region whose RT surface (gray dashed line) is tangent to $\Sigma$ at $Q_i$.

We can conclude that, for any area element on the RT surface of any static spherical region, the strength of the PEE flux passing through this area element is exactly $1/4G$.

Then we consider an arbitrary infinitesimal area element at any bulk point $Q$ with a random unit normal vector $n^\mu(Q)$. It is crucial to notice that, any area element can be embedded on a unique RT surface $\mathcal{E}_A$ of a spherical region $A$, where $\mathcal{E}_A$ is the hemisphere passing through $Q$ and normal to $n^\mu(Q)$ (see Fig. 6 for the cases of infinitesimal line segments in AdS$_3$). Then we can calculate the PEE flux of this area element by just collecting the threads emanating from the corresponding spherical region $A$. Remarkably, according to our discussion for the spherical regions, we conclude that

- *the strength of the PEE flux through any area element in the bulk is always $1/4G$.*

In other words, given any bulk point $Q$ and any direction $n^\mu(Q)$, we can determine a spherical region $A$ and calculate the strength of the PEE flux in the following way

$$\int_A \mathrm{d}^{d-1}\mathbf{x}|V_\mathbf{x}^\mu(Q)n_\mu(Q)| = \frac{1}{2}\int_{\partial\mathcal{M}} \mathrm{d}^{d-1}\mathbf{x}|V_\mathbf{x}^\mu(Q)n_\mu(Q)| = \frac{1}{4G}. \tag{22}$$

Here the factor $1/2$ comes from the fact that, the PEE threads emanating from $\bar{A}$ and passing through $Q$ are exactly those emanating from the spherical region $A$.

It is true that, an infinitesimal area element in the bulk can also be embedded on the RT surface $\mathcal{E}_A$ of a non-spherical region $A$. In this case, the classification for $\omega_{\mathcal{E}_A}(\mathbf{x},\mathbf{y})$ should change[2], and the calculation for the PEE flux passing through this area element is no-longer given by (21). The flux through an area element is independent from the way of embedding, and choosing spherical regions is just a trick to simplify the calculation.

Now we consider an arbitrary homologous surface $\Sigma_A$ in the AdS bulk homologous to any boundary region $A$, and divide $\Sigma_A$ into infinitesimal area elements. At this point, we are ready to prove our proposal (20) by applying the above conclusion to all the area elements on $\Sigma_A$. Straightforwardly, we get

$$\begin{aligned}S_A &= \frac{1}{4G}\min_{\Sigma_A}\int_{\Sigma_A}\mathrm{d}\Sigma_A\sqrt{h}\\ &= \min_{\Sigma_A}\frac{\mathrm{Area}[\Sigma_A]}{4G} = \frac{\mathrm{Area}[\mathcal{E}_A]}{4G},\end{aligned} \tag{23}$$

which is exactly the RT formula.

---

[2]For example, when $A$ is not a spherical region, we will have PEE threads with $\omega_{\mathcal{E}_A}(\mathbf{x},\mathbf{y}) > 1$. See Fig.4 for the case of a trip region.

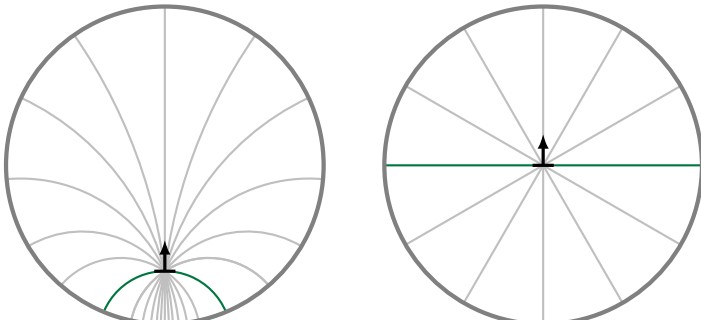

Figure 7: The area element $\mathrm{d}\Sigma$ and its direction is represented by the black arrow. The gray curves represent all the PEE threads that involve in the reconstruction of $\mathrm{d}\Sigma$, and the green curve represents the PEE thread that saturates the lower bound.

## 4 Bulk geometry reconstruction

Our goal is to interpret all the geometric quantities in terms of the boundary PEEs. We just interpreted the RT surfaces as the homologous surface that minimizes the flux of the PEE flow from $\mathcal{M}_A$ to $\mathcal{M}_{\bar{A}}$, and the minimized flux gives the area of the RT surface. In other words, we reconstructed all the RT surfaces via the class of PEE threads passing through it.

Similarly, we can reconstruct an arbitrary infinitesimal bulk co-dimension two surface $\mathrm{d}\Sigma$ from certain class of boundary two-point PEEs $\mathcal{I}(\mathbf{x}, \mathbf{y})$, whose PEE threads pass through $\mathrm{d}\Sigma$. According to our previous discussion, the class of PEE threads that passes $\mathrm{d}\Sigma$ are those emanating from a spherical region whose RT surface is tangent to $\mathrm{d}\Sigma$, and the flux of the PEE flow passing through $\mathrm{d}\Sigma$ gives Area($\mathrm{d}\Sigma$),

$$\frac{\text{Area}[\mathrm{d}\Sigma]}{4G} = \frac{1}{2} \int_{\partial\mathcal{M}} \mathrm{d}^{d-1}\mathbf{x} \int_{\partial\mathcal{M}} \mathrm{d}^{d-1}\mathbf{y} \, \omega_{\mathrm{d}\Sigma}(\mathbf{x}, \mathbf{y}) \mathcal{I}(\mathbf{x}, \mathbf{y}), \tag{24}$$

where $\omega_{\mathrm{d}\Sigma}(\mathbf{x}, \mathbf{y}) = 0$ or $1$ depending on whether the thread intersects with $\mathrm{d}\Sigma$. The PEE threads that do not intersect with $\mathrm{d}\Sigma$ will not participate its reconstruction.

Interestingly, given the position of a $\mathrm{d}\Sigma$, the scale $|\mathbf{x} - \mathbf{y}|$ of all the two-point PEEs $\mathcal{I}(\mathbf{x}, \mathbf{y})$ that participate the reconstruction of $\mathrm{d}\Sigma$, is lower bounded. For example, in Poincaré AdS$_{d+1}$, the scale of those two-point PEEs satisfy

$$|\mathbf{x} - \mathbf{y}| \geq 2z_0, \tag{25}$$

where $z_0$ is the $z$ coordinate of $\mathrm{d}\Sigma$. In other words, the two-point PEEs with $|\mathbf{x} - \mathbf{y}| < 2z_0$ will not contribute to the reconstruction of area elements deeper than $z_0$. See Fig. 7 for two examples in global AdS$_3$, where we show the sets of PEE threads that reconstruct two different area elements. In the left case, the $\mathrm{d}\Sigma$ is close to the boundary, hence the small scale PEEs contribute. In the right case, the $\mathrm{d}\Sigma$ is in the center of the AdS space and only the largest scale PEEs involve in its reconstruction.

We can also reconstruct any co-dimension two surface $\Sigma$ in the bulk, which does not need to be homologous to any boundary region. Because we can reconstruct all the infinitesimal area elements on the surface, see Fig. 6. Then all the PEEs that locally pass through $\Sigma$ will contribute to the reconstruction. And the area of $\Sigma$ will be reproduced by the flux locally passing through it, which is just

$$\frac{\text{Area}[\Sigma]}{4G} = \frac{1}{2} \int_{\partial\mathcal{M}} \mathrm{d}^{d-1}\mathbf{x} \int_{\partial\mathcal{M}} \mathrm{d}^{d-1}\mathbf{y} \, \omega_{\Sigma}(\mathbf{x}, \mathbf{y}) \mathcal{I}(\mathbf{x}, \mathbf{y}). \tag{26}$$

We can also consider the reconstruction of a lower dimensional object, i.e. a geometric object $\zeta$ of codimension-$(2 + n)$. The area of $\zeta$ is given by

$$\text{Area}[\zeta] = \int_\zeta \mathrm{d}\sigma_\zeta \sqrt{h_\zeta}, \tag{27}$$

where $\mathrm{d}\sigma_\zeta$ is the area element of $\zeta$, and $h_\zeta$ is the metric on $\zeta$. Using (22), we have

$$\frac{\text{Area}[\zeta]}{4G} = \frac{1}{2} \int_\zeta \mathrm{d}\sigma_\zeta \sqrt{h_\zeta} \int_{\partial\mathcal{M}} \mathrm{d}^{d-1}\mathbf{x} |V_{\mathbf{x}}^\mu(Q) n_\mu(Q)|, \tag{28}$$

where we let $Q$ be the location of $\mathrm{d}\sigma_\zeta$. There is an ambiguity on the direction of $n^\mu(Q)$ if $n \geq 1$, as there are $(n+1)$ directions that are orthogonal to $\mathrm{d}\sigma_\zeta$. Nevertheless, the integral in (22) is independent of the direction of $n^\mu$. When we integrate over $\zeta$, we take into account all the PEE threads that intersect with $\zeta$. Then (28) can be written on CFT side as

$$\frac{\text{Area}[\zeta]}{4G} = \frac{1}{2} \int_{\partial\mathcal{M}} \mathrm{d}^{d-1}\mathbf{x} \int_{\partial\mathcal{M}} \mathrm{d}^{d-1}\mathbf{y} \, \omega_\zeta(\mathbf{x},\mathbf{y}) \mathcal{I}(\mathbf{x},\mathbf{y}), \tag{29}$$

where $\omega_\zeta$ counts the number of times the PEE thread connecting $\mathbf{x}$ and $\mathbf{y}$ intersects with $\zeta$. Thus all the bulk geometric objects can be encoded in the boundary weight function $\omega_\zeta(\mathbf{x},\mathbf{y})$.

## 5 Discussion and conclusion

In summary, based on the PEE structure of a CFT and its geometrization scheme represented by all the geodesics anchored on the boundary, we obtain a network of geodesics which could be considered as the basic elements forming the AdS space in AdS/CFT. We show that the strength of the PEE flow at any bulk point in any direction is always $1/4G$. If we set $1/4G$ as the upper bound of the flow strength, then we can claim that the AdS space is full of PEE threads everywhere. Then any geometric quantity in AdS can be reconstructed by a set of boundary PEEs whose PEE threads passing through it.

We also provide a complete reformulation of the RT formula, which aims to identify the homologous surface $\Sigma_A$ with minimal PEE flux passing through it. The key of our scheme is that, we need to abandon the naive normalization property (3) of the PEE, which tells you to collect the PEE threads stretching between $A$ and $\bar{A}$. Instead, we should collect the PEE flux stretching between the two sides of the homologous surface $\Sigma_A$. This is partially inspired by the evaluation of the entanglement entropy of $\text{CFT}_2$ in the model of tensor networks, where the homologous path has the minimal number of cuts with the network. Here the PEE network is analogous to the tensor network, and the PEE flux through the homologous surface is akin to the number of cuts the surface intersects with the PEE network.

It is intriguing to view the PEE network as the tensor network which precisely captures the entanglement structure of the boundary CFT at large $c$ limit. Compared with previous tensor network toy models of gravity, the PEE network is a well-defined continuous network that naturally extends to higher dimensions. It will be interesting to add bulk degrees of freedom to the PEE network, to study the quantum correction and quantum error correction property of the PEE network. Extending or testing our scheme to more generic asymptotic AdS spacetimes, to the covariant configurations and to holographies beyond AdS/CFT are also worth exploring.

Our scheme gives a solution to an old puzzle of Casini and Huerta proposed in [60] (see also [41]). In that paper the authors consider the so-called extensive mutual information (EMI), which is the same as the PEE we studied, and use the normalization property of EMI to evaluate the entanglement entropy for an annulus in the vacuum of holographic $\text{CFT}_3$ that dual

to Poincaré AdS$_4$. The RT formula tells us that, the RT surface has a phase transition between two phases where the RT surface is connected or disconnected respectively. Nevertheless, if we naively apply (3) to compute the entanglement entropy, there is no such phase transition. Hence the authors of [66] claimed that, the EMI does not exist in holographic CFTs. While our scheme reproduces the results of the RT formula perfectly, by considering the weight of the two-point PEEs. More specifically, in our scheme PEE threads across the annulus (i.e. PEE threads stretching between the region closed by the annulus and the region outside the annulus) could contribute to the entanglement entropy of the annulus in the disconnected phase, which goes beyond the naive normalization property (3).

Our scheme can be used to reconstruct generic bulk geometric quantities in general dimensions, which we think is a big progress based on the previous schemes we reviewed. So far our discussion is confined in a static time slice of Pure AdS$_{d+1}$. In order to generalize our scheme to generic spacetimes, we need to analysis the PEE structure of the boundary states and geometrize the two-point PEEs into bulk geodesics anchored on the boundary. For spacetimes with a black hole in the bulk, there will be PEE threads passing through the black hole horizon. Also, it will be very important to generalize it to the covariant configurations. We leave these for future investigations.

## Acknowledgements

Y. Lu is supported by the National Natural Science Foundation of China under Grant No.12247161, the NSFC Research Fund for International Scientists (Grant No. 12250410250) and China Postdoctoral Science Foundation under Grant No.2022TQ0140. J. Lin is supported by the National Natural Science Foundation of China under Grant No.12247117, No.12247103. Q. Wen would like to thank Bartlomiej Czech, Veronika E. Hubeny, Ling-Yan Hung, Huajia Wang and Zhenbin Yang for helpful discussions.

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
