# Peer review of "Partial entanglement network and bulk geometry reconstruction in AdS/CFT"

_SciPost Physics_

## Round 1 · Referee Report · Anonymous · 2024-6-27

Weaknesses

1. Contains essentially the same results as ref. [1]. Nothing substantial is added to merit a separate publication.

2. The proposed thread prescription depends on the bulk geometry and state. It only seems to apply to spheres in vacuum AdS.

3. The 'reformulation' of the RT prescription is wrong in general.

4. The boundedness condition is not checked or used, which is essential to the max flow program.

Report

The manuscript 'Partial Entanglement Network and Bulk Geometry Reconstruction in AdS/CFT' seeks to offer a novel characterization of bulk geometry through the concept of PPE bit threads. However, the paper predominantly comprises an extensive review of [1], with the original contributions confined to section 4, which spans only one page. Consequently, the manuscript does not present enough new material to justify publication. Additionally, the paper is deficient in detail and contains several erroneous claims, some of which are inherited from [1]. I will elaborate on these issues below.

Firstly, the two-point PPE I(x,y) highly depends on the state. The one used in this paper works for the AdS vacuum, but does not easily generalize to other geometries or excited states. Since, I(x,y) is linked to a geodesic, it effectively encodes a two-point function of the boundary theory in the large conformal dimension limit. However, in general geometries and states, boundary-anchored geodesics span a region in the bulk that is parametrically smaller than the region spanned by minimal surfaces. Therefore, I do not believe the construction presented in this paper is universally applicable. A simple case illustrating this limitation is the presence of a black hole. In such a scenario, some of the threads must necessarily reach the horizon, and thus cannot be encoded in the boundary two-point function.

Secondly, even in the AdS vacuum, the analysis for multiple regions appears to have several flaws. First, the 'reformulation' is not entirely independent of the RT prescription. The RT surfaces are required to determine the weights w_i, which must be manually inserted into the threads program. In a true reformulation, these weights should be dynamic variables. Second, the assigned weights in different phases (connected/disconnected) imply that entanglement sometimes accounts for threads connecting two points within region A (or its complement), which is contrary to what the max flow program actually does. According to the original bit threads paper (ref. [23] in the article), S_A​ should only count the number of threads connecting region A and its complement, representing the EPR pairs that can be distilled from the state. Threads connecting two points within A (or its complement) should not affect the entropy.

What does this imply? While Equation (17) in the paper is correct in terms of entropies (or areas), the proposed thread version of this equation will NOT solve the max flow program. It will result in a divergenceless vector field that yields the correct entropy, but its norm will not necessarily be bounded. Furthermore, the authors do not attempt to prove the boundedness of their proposed thread constructions in the general case. They also fail to provide an explicit expression for V^mu in the case of multiple regions (not even provided in ref. [1]), which could be used to explicitly check whether the norm bound is violated or satisfied in that particular case.

Based on the issues outlined above, I believe the fundamental premise of the paper is flawed and does not meet the standards for publication in SciPost.

Recommendation

Reject

  • validity: poor
  • significance: poor
  • originality: poor
  • clarity: poor
  • formatting: reasonable
  • grammar: good

Author:  Qiang Wen  on 2024-07-07  [id 4604]

(in reply to Report 1 on 2024-06-27)
Category:
answer to question
objection
suggestion for further work

First of all, we would like to thank the referee for the detailed report. Nevertheless, we are surprised when we get this report. It seems that the referee has missed the main points of this paper and ref. [1], and underestimate our work for the failure in reproducing the Bit thread configurations for non-spherical boundary regions. In the following we give clarifications including: 1) the difference between our setup and the Bit threads; 2) the difference between this submission and the ref. [1]. Also we would like to add a section to discussion the equivalence between our formula and the Crofton formula in AdS. We hope the referee can give us a chance to resubmit.

1) The PEE threads are bulk geodesics anchored on the boundary, and the density of the threads are determined by the boundary PEE structure, i.e. $\mathcal{I}(x,y)$. Although the setup of PEE threads was partially inspired by the Bit thread configurations, the way they describe entanglement structure of a holographic state is essentially different, and there is no reason to require the PEE network configuration should reproduce reasonable Bit thread configurations.

a. For a given state, the PEE structure and the bulk network of the PEE threads are determined. While the Bit thread configurations are highly degenerate even when the state and boundary region under study are fixed.
b. Although, like the Bit threads, we used divergenceless vector fields to describe the PEE threads emanating from any boundary point, we didn’t set upper bound for the norm of the vector field. More importantly, the PEE threads are un-oriented lines.
c. Given a boundary region A, the quantities that capture $S_A$ in the two configurations are essentially different. In the PEE thread configurations, we consider all possible homologous surface $\Sigma_A$ in the bulk and optimize the number of times it intersect with the network of PEE threads. While in the Bit thread configurations, the bit threads are oriented lines and we need to optimize the flux of the bit threads from A to $\bar{A}$. This difference is manifest by considering a single thread that intersect with a homologous surface twice. If it is a bit thread, then its contribution to $S_A$ is zero, while for a PEE thread, its contribution to $S_A$ doubles.

For the special case of the RT surface for spherical regions in vacuum CFT, the two quantities coincide as all the PEE threads intersect with the RT surface at most once. In this case, counting the number of intersections is equivalent to evaluating the flux of PEE threads from A to $\bar{A}$, so we can reproduce a bit thread flow from the PEE threads in ref.[1]. But this coincidence does not hold for non-spherical regions and disconnected regions. In summary, we should not expect to reproduce Bit thread flow configurations from the PEE network except for the spherical regions.

2) In ref.[1] we defined the PEE threads for Poincare AdS in general dimensions, and calculate the strength of the PEE flow on the RT surfaces of spherical regions and find it to be 1/4G when assuming that the PEE threads are pointing from A to $\bar{A}$. On a time slice of AdS3, we proposed that the RT surface should minimize the number of intersections between the homologous surface and the PEE threads, and give some tests for this proposal. But in ref.[1] this proposal is only confined in a time slice of AdS3 and it is not proved.

The main problem remains in ref.[1] is that, we do not know how to compute the number of intersections for general surfaces that are not the RT surfaces for spherical regions (see the appendix of ref.[1] for an unsuccessful trail). In this submission we solved it in a surprisingly simple way. It turns out that, the answer is simple and all the calculation we need is in the case of spherical regions carried out in ref.[1]. Consider an infinitesimal area element at the bulk point Q and with the normal direction $\vec{n}$, there exists an unique hemi-sphere in the bulk that is tangent to this area element. This hemi-sphere is the RT surface of a spherical region A. When we calculate the number of intersections between this area element and the bulk PEE threads, we need to avoid double counting. It is interesting to find that, one can avoid double counting by only consider the threads emanating from A (see fig. 5). Based on the computation for spherical regions, the number of intersections between the area element and bulk PEE threads is exactly the area of the area element multiplied by 1/4G.

Since the above result holds for any bulk area element, one can immediately get the number of intersections for any bulk surface, which is the area of the surface multiplied by 1/4G. So for any boundary region, when we optimize the number of intersections for all possible homologous surfaces, we are actually optimizing the area of the homologous surfaces, which reproduce the RT formula straightforwardly.

In summary, in ref.[1] we gave the setup of PEE threads and a proposal to reproduce the RT formula in AdS3, and in this submission we generalize this proposal to higher dimensions and prove it. The referee was right that our discussion is currently confined in the vacuum state of holographic CFTs. But the reformulation of the RT formula works for generic boundary regions, including the disconnected ones. And our geometric reconstruction goes beyond the RT surfaces.

If the referee give us a chance for resubmission, we will add the discussion on the equivalence between our result (eq. (26)) and the Crofton formula, which is noticed to us very recently. We found that the boundary PEE structure coincide with the measure for the kinematic space (which can be understand as the space of bulk geodesics) for AdS. Since the Crofton formula works for more generic geometry backgrounds, we may try to extend our discussion for geometries beyond pure AdS in the future.

In the following we will answer some of the questions from the referee.

a. Referee: However, the paper predominantly comprises an extensive review of [1], with the original contributions confined to section 4, which spans only one page.

Reply: Section 3 and 4 are new results

b. Referee: However, in general geometries and states, boundary-anchored geodesics span a region in the bulk that is parametrically smaller than the region spanned by minimal surfaces. Therefore, I do not believe the construction presented in this paper is universally applicable. A simple case illustrating this limitation is the presence of a black hole. In such a scenario, some of the threads must necessarily reach the horizon, and thus cannot be encoded in the boundary two-point function.

Reply: When there is black hole in the bulk, the boundary state is mixed, and there exists PEE threads emanating from the boundary and enters the black hole interior. In this case one can not use the two boundary endpoints to characterize all the PEE threads. We may consider the eternal black hole where the PEE between the two boundaries may help us determine the density of threads stretching between the two boundaries. Another way to understand such configurations is using the Crofton formula and interpreting all the bulk geodesics in terms of the two-point PEEs. These not only include the PEE between boundary points, but also the PEEs between boundary and the black hole interior. We leave this for future investigation.

c. Referee: The RT surfaces are required to determine the weights w_i, which must be manually inserted into the threads program. In a true reformulation, these weights should be dynamic variables.

Reply: The eq. (19) and eq. (20) are calculating the same thing, i.e. the number of intersections between the $\Sigma_A$ and the bulk PEE threads. In (19) it was count thread by thread which is formidable to calculate. In (20) it was count along the surface, which fortunately turns out to be the area of $\Sigma_A$ multiplied by 1/4G. Here the PEE threads are fixed, and the optimization is among all possible $\Sigma_A$.

d. Referee: Secondly, even in the AdS vacuum, the analysis for multiple regions appears to have several flaws. First, the 'reformulation' is not entirely independent of the RT prescription.

Reply: The reformulation is independent of the RT prescription as the optimization is among all possible $\Sigma_A$.

e. Referee: Second, the assigned weights in different phases (connected/disconnected) imply that entanglement sometimes accounts for threads connecting two points within region A (or its complement), which is contrary to what the max flow program actually does. According to the original bit threads paper (ref. [23] in the article), S_A should only count the number of threads connecting region A and its complement, representing the EPR pairs that can be distilled from the state. Threads connecting two points within A (or its complement) should not affect the entropy. What does this imply?

Reply: The referee was right about the setup about bit threads, this is not our setup. When we go beyond the spherical regions, threads connecting points within A (or $\bar{A}$) do contribute to $S_A$. This means large scale entanglement within A could contribute to $S_A$. We would like to take this as a new understanding about the entanglement structure which was omitted before, and hope to understand it better in the future.

f. Referee: While Equation (17) in the paper is correct in terms of entropies (or areas), the proposed thread version of this equation will NOT solve the max flow program. It will result in a divergenceless vector field that yields the correct entropy, but its norm will not necessarily be bounded. Furthermore, the authors do not attempt to prove the boundedness of their proposed thread constructions in the general case. They also fail to provide an explicit expression for V^mu in the case of multiple regions (not even provided in ref. [1]), which could be used to explicitly check whether the norm bound is violated or satisfied in that particular case.

Reply: We do not expect to reproduce any bit thread configurations that solve the max flow program for non-spherical regions. As we have mentioned at the beginning that, the setups of bit threads and PEE threads are essentially different.

---

## Editorial Decision

in_refereeing